# The Effects of Physical Activity and Diet Interventions on Body Mass Index in Latin American Children and Adolescents: A Systematic Review and Meta-Analysis

**DOI:** 10.3390/nu12051378

**Published:** 2020-05-12

**Authors:** Andrés Godoy-Cumillaf, Paola Fuentes-Merino, Armando Díaz-González, Judith Jiménez-Díaz, Vicente Martínez-Vizcaíno, Celia Álvarez-Bueno, Iván Cavero-Redondo

**Affiliations:** 1Grupo de investigación en Educación Física, Salud y Calidad de Vida, Facultad de Educación, Universidad Autónoma de Chile, Temuco 4810101, Chile; andres.godoy@uautonoma.cl (A.G.-C.); paola.fuentes@uautonoma.cl (P.F.-M.); 2Secretária de Esportes Taubaté, São Paulo 12030-670, Brazil; diazgonzalez73@gmail.com; 3Escuela de Educación Física y Deportes, Universidad de Costa Rica, San Jose 11-501-2060, Costa Rica; judith.jimenez_d@ucr.ac.cr; 4Health and Social research, Universidad de Castilla-La Mancha, 16071 Cuenca, Spain; vicente.martinez@uclm.es (V.M.-V.); ivan.cavero@uclm.es (I.C.-R.); 5Universidad Politécnica y Artística del Paraguay, Asunción 001518, Paraguay

**Keywords:** obesity, overweight, BMI, nutrition, weight status

## Abstract

The objective of this systematic review and meta-analysis is to compare the effect of physical activity only with that of physical activity plus diet interventions on body mass index (BMI) in Latin American children and adolescents. We searched the Medline, Embase, Scopus, Web of Science, and Scielo databases from their inception until March 2020, including studies examining the effect of physical activity or physical activity plus diet interventions on BMI in children and adolescents and based on data from intervention studies. The DerSimonian and Laird method was used to compute a pooled standardized mean difference for BMI in terms of effect size (ES) and respective 95% confidence intervals (CIs). Eighteen studies were included. Analyses were performed based on intervention (four studies were included for physical activity only and four studies were included for physical activity plus diet). In the analysis of physical activity only versus control, there was no effect on BMI (ES = 0.00; 95% CI −0.17–0.17, I^2^ = 0.0%; *p* = 0.443). In the analysis of physical activity plus diet versus control, there was a decrease in BMI in favour of the intervention group (ES = −0.28; 95% CI −0.42–−0.14, I^2^ = 74.5%; *p* = 0.001). When ES was estimated considering only the effect in intervention groups, there was no evidence of a decrease in BMI (ES = −0.17; 95% CI −0.44–0.11, I^2^ = 84.5%; *p* < 0.001) for physical activity only (eight studies). However, there was a statistically significant decrease in BMI (ES = −0.30; 95% CI −0.50–0.11, I^2^ = 95.8%; *p* < 0.001) for physical activity plus diet (ten studies). Some limitations of this review could compromise our results, but the main limitation that should be stated is the quality of the studies (mainly medium/moderate), especially as physical activity and diet interventions cannot be blinded, compromising the quality of these studies. In summary, this meta-analysis offers evidence that physical activity plus diet interventions produced a reduction in BMI in Latin American children and adolescents, but physical activity only interventions did not.

## 1. Introduction

The levels of overweight and obesity in children and adolescents in Latin America have been rising significantly over the last few years, becoming an important public health problem [1,2]. It has been estimated that between 20% and 25% of Latin American children and adolescents are overweight or obese, as established according to body mass index (BMI) [3]. Overweight and obesity affect children’s health in both the short and long term, increasing the prevalence of muscle pain, lesions, or fractures [4], obstructive sleep apnoea [5], and worse self-esteem and quality of life [6]. Furthermore, overweight and obesity increase the risk of cardiovascular disease [7], high blood pressure [8], some types of cancer [9], and all causes of premature death [10], factors which are maintained and increased during adulthood [11,12].

BMI has commonly been used to estimate the prevalence of overweight or obesity in childhood and adolescence [13,14,15], as it is a good indicator of general adiposity [16,17,18]. Despite its value as an indicator, it should be cautiously interpreted because it does not accurately reflect changes in adiposity in children and adolescents [19], especially in boys who are underweight [20]. However, it is still one of the most sensible and widely used methods to measure adiposity in children and adolescents [14].

Physical activity is recommended to prevent overweight and obesity in children and adolescents [21,22,23], as it plays an important role in reducing BMI and health risk factors [24,25]. Furthermore, a healthy diet is one of the basic conditions for a healthy population which is prepared to prevent most highly prevalent diseases, such as obesity, diabetes, hypertension, or high levels of cholesterol [26].

Previous studies analyzed the effect of physical activity interventions versus interventions that combine physical activity plus diet on BMI, showing a higher effect in the physical activity plus diet interventions groups [27,28]. In Latin America, four previous systematic reviews on interventions aimed at increasing physical activity in children have been performed, some of which were combined with diet intervention [29,30,31,32]. These systematic reviews showed that when physical activity interventions are combined with a nutritional component, they are more effective in producing favourable changes in adiposity (i.e., body parameters, body composition, and BMI). However, none of the four previously mentioned reviews performed meta-analyses to compare physical activity versus physical activity plus diet versus control groups. Thus, it is a complex task to evaluate whether the physical activity and diet interventions in Latin America have produced an impact on BMI, or in which population (underweight, normal, overweight, or obese), or what type of physical activity intervention (complemented with diet interventions) had the best results. For these reasons, the objective of this systematic review and meta-analysis is to compare the effect of physical activity only with that of physical activity plus diet interventions on BMI in Latin American children and adolescents.

## 2. Materials and Methods

This study was developed following the Preferred Reporting Items for Systematic Reviews and Meta-Analyses (PRISMA) guidelines [33] and the suggestions of the Cochrane Collaboration Handbook [34]. PROSPERO (CRD42019077702) was used to register the systematic review and meta-analysis. Its protocol has been previously published in [35].

### 2.1. Search Strategy

A literature review was conducted independently by two researchers in the following databases: Medline (via PubMed), Embase, Scopus, Web of Science, and Scielo. Eligible articles were those which examined the effect of physical activity or physical activity plus diet interventions on BMI in children and adolescents and which were based on data from intervention studies. The search strategy was based on the terms described in Table 1. The citations in the articles considered to be eligible for the systematic review were checked to complement the literature search. 

### 2.2. Study Selection

To be included, studies needed to meet the following criteria: (1) participants from a Latin American country; (2) participants aged between 4–18 years old; (3) addressing physical activity interventions (physical endurance, sports, or alternative exercise such as games, dancing, optimised physical education classes; including or not diet intervention); (4) randomised controlled trial (RCT), non-randomised controlled trial (non-RCT), or single-arm pre-post study; (5) showing BMI before and after the intervention; (6) studies written in Spanish, English, or Portuguese; and (7) published prior to March 2020.

Additionally, the exclusion criteria were: (1) participants younger than four years or older than 18 years; or (2) self-reported BMI.

After exclusion of duplicate studies, titles and abstracts of the included articles were evaluated by two researchers to identify eligible studies. Abstracts partially fulfilling the inclusion/exclusion criteria (i.e., which did not provide enough information) were evaluated through reading of the complete text. Subsequently, the same two researchers examined the included and excluded studies to verify the reason for each decision. A third researcher made the final decision by analyzing and resolving the inconsistencies between the two researchers, strictly based on the inclusion/exclusion criteria.

### 2.3. Data Extraction and Quality Assessment

The following data were extracted from the original articles: (1) year of publication; (2) country; (3) study design; (4) age of participants; (5) sample size; (6) type of population (normal, overweight, or obese); (7) type of physical activity intervention; (8) characteristics of physical activity intervention; (9) type of diet intervention; and (10) characteristics of the diet intervention. 

The methodological quality of randomised controlled trials (RCTs) was assessed using the Cochrane Collaboration tool for assessing risk of bias (RoB2) [36], according to the Cochrane Collaboration Handbook recommendations. RoB2 evaluates the risk of bias according to six domains: randomisation process, assignment to intervention, adhering to intervention, missing outcome data, measurement of the outcome, and selection of the reported result. Overall bias was considered as one of the following: “low risk of bias” (the study is classified as “low risk” in all domains), “some concerns” (at least one domain with a rating of “some concern”), and “high risk of bias” (at least one domain with “high risk” or several domains with “some concerns”, considered critical to the result validity).

The methodological quality of the non-RCTs was assessed using the quality assessment tool for quantitative studies [37]. This tool assesses six domains: selection bias, study design, confounders, blinding, data collection method, and withdrawals and drop-outs. Each domain can be scored as strong, moderate, or weak. If no domain is qualified as weak, a study is scored as strong; it is scored as moderate if one domain is qualified as weak; or it is scored as weak if two or more domains are qualified as weak.

The quality assessment was independently done by two reviewers (A.G.-C and A.D.-G), and inconsistencies were resolved by consensus or with the participation of a third researcher (I.C.-R).

### 2.4. Statistical Analysis

The DerSimonian and Laird method was used to compute a pooled estimate of effect size (ES) and respective 95% confidence intervals (CIs). Analyses were performed based on the type of intervention (physical activity only or physical activity plus diet). When studies were RCTs, a standardised mean difference value was calculated for BMI using Cohen’s d index as the ES statistic [38]. In addition, Cohen’s d index (as the ES statistic) was used to estimate pre-post physical activity intervention changes in BMI. Cohen’s d values around 0.2 were considered to indicate a weak effect, values around 0.5 were considered to indicate a moderate effect, values around 0.8 were considered to indicate a strong effect, and values larger than 1.0 were considered to indicate a very strong effect. The heterogeneity of the results across the studies was evaluated using the I^2^ statistic, with an I^2^ value between 0% to 30% heterogeneity representing “not important”, 30–50% representing moderate heterogeneity, 50–80% representing substantial heterogeneity, and 80–100% representing considerable heterogeneity. The corresponding p values and ES 95% for I^2^ were also considered [33]. Furthermore, a sensitivity analyses were performed using the restricted maximum likelihood (REML) method to estimate the heterogeneity variance [39]. Additionally, when studies included two intervention groups, their data were analysed as independent samples. 

To assess the robustness of summary estimates and to detect whether any particular study accounted for a large proportion of heterogeneity, sensitivity analyses were conducted. Performance of sub-group analyses were based on sex, study design (RCT, non-RCT, or single-arm pre-post study), and weight status (overweight/obese or general population). Both analyses were defined as post-hoc.

Finally, publication bias was evaluated through visual inspection of funnel plots, as well as using the method proposed by Egger [40]. Statistical analyses were performed using the STATA® SE software, version 15 (StataCorp, College Station, TX, USA).

## 3. Results

### 3.1. Systematic Review

The systematic review and meta-analysis flow diagram is presented in Figure 1. Eighteen studies published between 2006 and 2016 were included. Six studies were performed in Chile, five in Brazil, five in Mexico, and two in Colombia. Ten of the studies only measured subjects in the intervention group, while the remaining eight studies also had a control group. However, only six of the aforementioned groups were randomised studies. 

The sample sizes ranged from nine to 2527. In eight studies, the participants were overweight and/or obese, while in the other 10 studies, the weight status of participants was not specified (Table 2). Regarding the physical activity interventions, the length of intervention ranged from two to 36 months, including different types of exercise, such as increment in the number of physical education classes, recreational activities during recess or school breaks, recreational sports, dance, games, hikes, relaxation, and even strength and flexibility training. The duration of the sessions was between 20 and 120 min, with frequency of 2–7 days a week.

Ten of the interventions were complemented with diet intervention, which had the objective of modifying diet or teaching about healthy diet and nutrition. Diet interventions included diet intervention, nutritional education, nutritional counselling, and educational group meetings.

Additionally, 13 studies were conducted in the school, two in physical activity laboratories at a university, two in clinical settings, and one outdoors (walking), through counselling on how to increase physical activity.

### 3.2. Quality Assessment

According to the assessment using the RoB2 [36] tool for the risk of bias in randomised studies and the tool for the assessment of quantitative studies [37] (Table 3), 27.8% of the studies had a high risk of bias, 61.1% had moderate risk, and 11.1% had a low risk.

### 3.3. Meta-Analysis

For the analysis of physical activity interventions versus control, the ES on BMI was 0.00 (95% CI −0.17–0.17) with no heterogeneity (I^2^ = 0.0%; *p* = 0.443). Furthermore, for the analysis of physical activity plus diet versus control, the ES on BMI was −0.28 (95% CI −0.42–−0.14) with substantial heterogeneity (I^2^ = 74.5%; *p* = 0.001; see Figure 2). After using REML method, ES and heterogeneity was not modified for physical activity interventions versus control (ES = 0.00; 95% CI −0.17–0.17; I^2^ = 0.0%; *p* = 0.443) and slightly modified for physical activity plus diet versus control (ES = 0.28; 95% CI −0.42– −0.13; I^2^ = 76.7%; *p* = 0.001).

Additionally, when ES was estimated considering only the effect in intervention groups, the ES on BMI was −0.17 (95% CI −0.44–0.11) for physical activity; and the ES on BMI was −0.30 (95% CI −0.50–−0.11) for physical activity plus diet, both with substantial heterogeneity (I^2^ = 84.5%; *p* < 0.001 and I^2^ = 95.8%; *p* < 0.001, respectively; see Figure 3). After using the REML method, ES and heterogeneity were rather modified for physical activity interventions versus control (ES = −0.36; 95% CI −1.07–0.36; I^2^ = 98.0%; *p* < 0.001) and slightly modified for physical activity plus diet versus control (ES = −0.37; 95% CI −0.79–0.05; I^2^ = 99.2%; *p* < 0.001).

### 3.4. Sensitivity Analysis

The pooled ES estimates were not significantly modified in magnitude or direction when individual study data were removed from the analysis one at a time, both for the analysis of intervention versus control and for pre-post intervention.

### 3.5. Sub-Group Analysis

Sub-group analysis based on sex showed that when both analyses (i.e., intervention versus control and pre-post intervention) were performed separately for boys and girls, there were no effects. Additionally, in the sub-group analysis based on study design, the effect was modified for non-RCT study design (ES = 0.22; 95% CI 0.15–0.29) for physical activity plus diet pre-post intervention. Finally, when weight status was analysed, the effect was modified in the general population (ES = 0.08; 95% CI −0.01–0.16; see Table 4).

### 3.6. Publication Bias

Evidence of publication bias was found by funnel plot asymmetry and the Egger test for physical activity plus diet pre-post intervention effects (*p* = 0.007).

## 4. Discussion

This systematic review and meta-analysis aimed to compare the effect of physical activity only with that of physical activity plus diet interventions on the BMI of Latin American children and adolescents. Our findings showed that physical activity combined with diet interventions was effective at reducing BMI in Latin American children/adolescents. Furthermore, the findings showed that physical activity plus diet interventions were more effective at reducing BMI among overweight and obese participants when the study was designed as an RCT and when girls and boys were analysed together. 

Regarding the effect of physical activity plus diet interventions in the treatment of overweight and obesity, our findings agree with previous investigations that studied non-Latin American populations, which reported an association between physical activity and BMI, as well as such interventions being an efficient way of lowering the percentage of adipose tissue [24,59,60,61,62,63]. Regarding the participants in the interventions, those who were overweight and obese achieved a larger reduction of BMI in comparison to participants with a normal weight, in agreement with previous evidence [25,64]. Physical activity and diet interventions help children who are overweight or obese to expend more energy than usual, causing a reduction of excess adipose tissue. However, this arises to a lesser degree in those with a normal weight, as they have less adipose tissue.

Our results on the effect of physical activity plus diet agree with previous meta-analyses that have analyzed non-Latin American populations [65,66], highlighting that physical activity is one of the central elements of weight loss. However, when combined with diet intervention, the reduction ranged from 3.2% to 20% more, underscoring that the best results are achieved when calories are restricted. All this confirms the necessity of designing interventions which combine physical activity with a nutritional component, as was done in three of the studies included in this meta-analysis, which concluded that subjects that included physical activity plus diet programs proved to be more efficient in decreasing BMI values in children and adolescents [42,53,56].

The significant results found in the reduction of BMI by combining physical activity with diet are important from a clinical point of view, as evidence has been provided that will help to prevent and treat overweight and obesity in children and adolescents, and which may also reduce the risk of acquiring metabolic and cardiovascular diseases, consequently reducing the economic expenditure on health that is currently generated by childhood overweight and obesity [67].

Our results demonstrate that boys and girls had similar reductions in their values of BMI, in agreement with previous findings [68,69,70,71]. This finding demonstrates that when the characteristics of the intervention are the same for both sexes, there are no significant differences between them in terms of weight loss. 

When individuals participated in randomised interventions, the effect size was greater than that when the participants were not randomly assigned. This design characteristic agrees with what was found in a previous meta-analysis that only included RCTs, reporting significant effects on BMI reduction [72]. Furthermore, a previous study which compiled the results of RCT and non-RCT studies showed significant reductions in the former and slight reductions in the latter [73]. Our findings highlight the need to standardise the design of the interventions, in order to obtain better results. It should be pointed out that, as a product of the poor quality of the included studies, the findings of this meta-analysis should be cautiously considered. 

Among the relevant aspects of this systematic review is that the majority of the studies analysed were conducted in schools, which is a good setting, as previous evidence has shown [29,72,74,75]. Schools provide greater support for improving both nutritional habits and physical activity while, at the same time, integrating the family, which causes the impact of the intervention to be sustainable over time.

The presence of overweight or obesity during childhood and adolescence is related to a higher risk of tracking this condition into adulthood [76], which leads to a higher probability of premature death [10]. Taking into account that currently 58% of Latin Americans are overweight and 23% are obese [23,77], this is a concern that makes the implementation of a greater number of weight management interventions in children and adolescents in Latin America imperative, which could be designed by taking into consideration the results of the present investigation.

The limitations of this review that could compromise its results should be stated. First, a potential source of bias could come from data extraction being non-blinded. Second, the studies were of medium quality overall. Third, the interventions were heterogeneous regarding type, length, and intensity, making the size of some groups very small for sub-group analysis. Fourth, the studies did not assess the daily physical activity of subjects performed outside of the interventions (either by questionnaire or accelerometer); thus, this confounding effect outside of the interventions could not be controlled. Fifth, given that a limited number of studies included in this meta-analysis assessed the BMI Z-score, it was not possible to perform analyses based on this indicator. Therefore, it is suggested that, in future studies performed in Latin American children, the BMI Z-score should be incorporated, as this indicator reflects the changes produced by growth and maturation in more detail. Sixth, there was publication bias on physical activity plus diet pre-post intervention effects, due to the lack of studies with low ES and high sample size.

## 5. Conclusions

To summarise, this meta-analysis offers evidence that physical activity plus diet interventions produced a reduction in BMI, but physical activity only interventions did not. Furthermore, the effect on BMI was higher when the interventions were performed with overweight or obese participants. Based on these positive findings, it is necessary to implement more physical activity plus diet interventions in Latin America, in order to help in reducing the high levels of overweight and obesity that are found in this region.

## Figures and Tables

**Figure 1 nutrients-12-01378-f001:**
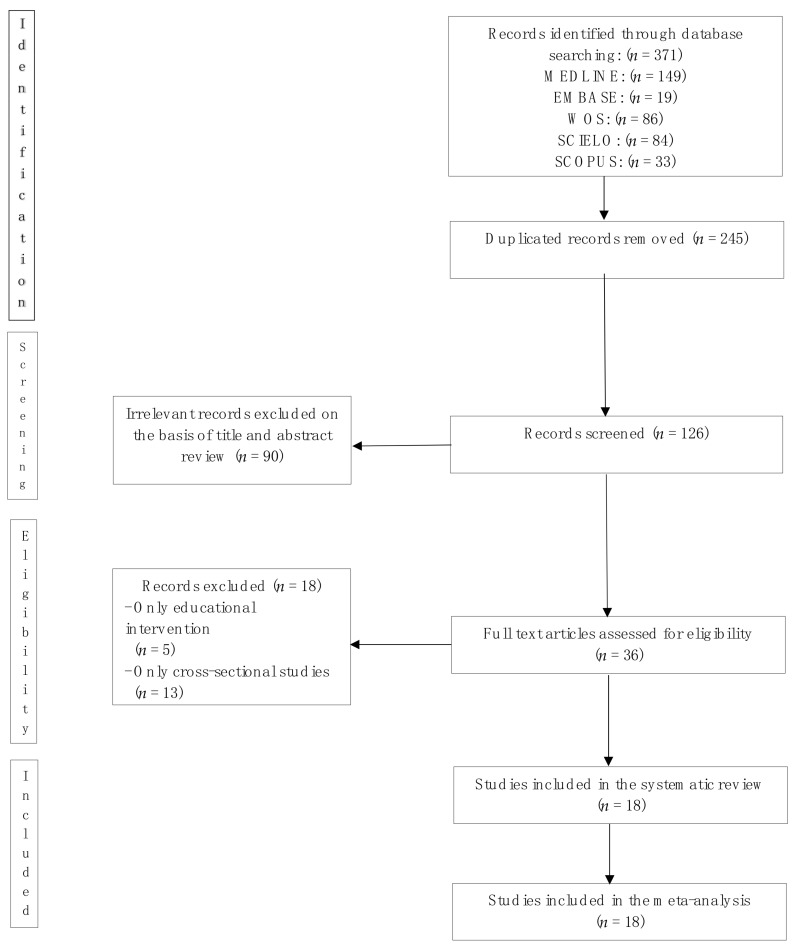
Preferred Reporting Items for Systematic Reviews and Meta-Analyses (PRISMA) flow diagram of identification, screening, eligibility, and inclusion of studies.

**Figure 2 nutrients-12-01378-f002:**
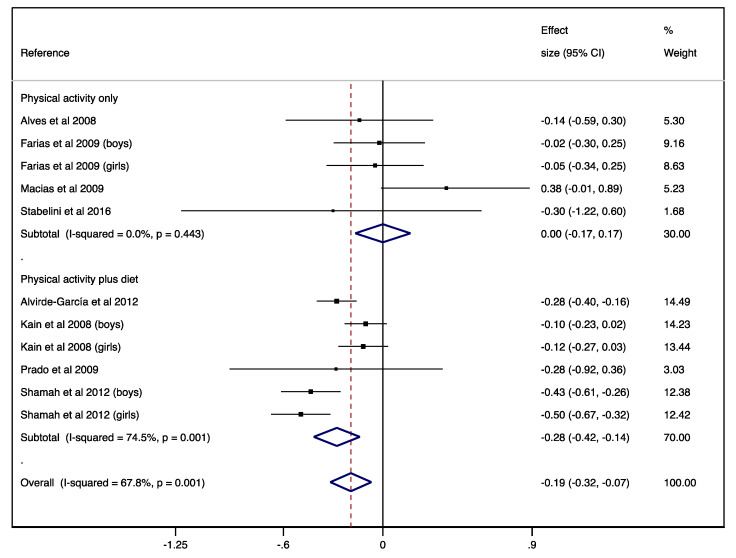
Forest plot for the randomised controlled trial studies. Diamond symbols represent the pooled effect size and 95% confidence interval; the lines with a solid square represent the effect size for each study and 95% confidence interval.

**Figure 3 nutrients-12-01378-f003:**
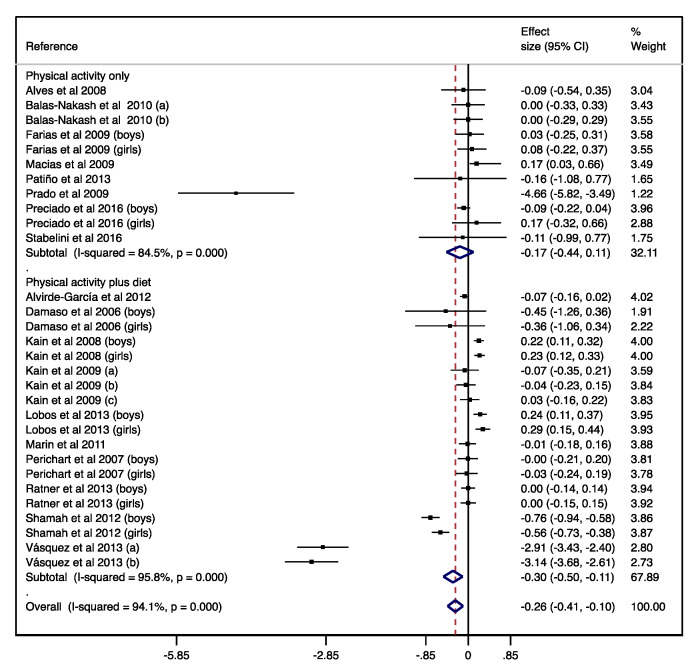
Forest plot for the single-arm pre-post studies. Diamond symbols represent the pooled effect size and 95% confidence interval; the lines with a solid square represent the effect size for each study and 95% confidence interval.

**Table 1 nutrients-12-01378-t001:** Search strategy for the MEDLINE database.

Search Set	Medline	Search Set	Medline
#1	physical activity [tw]	#18	paediatric obesity [mh]
#2	physical exercise [tw]	#19	adolescents [tiab]
#3	physical performance [tw]	#20	youth [tiab]
#4	exercise program* [tiab]	#21	14 OR 15 OR 16 OR 17 OR 18 OR 19 OR 20
#5	physical fitness [mh]	#22	underweight [tw]
#6	exercise [mh]	#23	normal weight [tw]
#7	1 OR 2 OR 3 OR 4 OR 5 OR 6	#24	overweight [tw]
#8	training program* [tiab]	#25	BMI [tiab]
#9	prevention program* [tiab]	#26	obesity [tiab]
#10	intervention program* [tiab]	#27	body mass index [mh]
#11	promotion [tw]	#28	22 OR 23 OR 24 OR 25 OR 26 OR 27
#12	strategy [tw]	#29	studies [tiab]
#13	8 OR 9 OR 10 OR 11 OR 12	#30	randomised controlled trials [tiab]
#14	childhood obesity [tiab]	#31	RCT [tiab]
#15	obese children [tiab]	#32	non-randomised experimental [tiab]
#16	obesity prevention [tiab]	#33	single-arm pre-post [tiab]
#17	obesity review [tiab]	#34	29 OR 30 OR 31 OR 32 OR 33
		#35	7 AND 13 AND 21 AND 28 AND 34

mh: MeSH terms; tiab: title/abstract; tw: text word. * is used to search for or all terms that begin with a word (e.g., “training program*” would return results such as “training programs” and “training programme”).

**Table 2 nutrients-12-01378-t002:** Characteristics of studies included in the systematic review and/or meta-analysis.

Population Characteristics						Intervention Characteristics			
Reference	Country	Study Design	Age Distribution	Sample Size	Type of Population	Physical Activity Intervention	Physical Activity Characteristics	Diet Intervention	Diet Characteristics
Alves et al. 2008 [41]	Brazil	RCT	5–10	78	Overweight and obese	In school, RS	150 min/week, 24 weeks total	-	-
Alvirde-García et al. 2013 [42]	Mexico	RCT	9 (average)	1224	General	In school, LA	Not specified	Diet intervention and nutritional education	Intervening in the food provided, informative texts provided to students
Balas-Nakash el al. 2010 [43]	Mexico	Pre-post intervention	8–12	319	General	In school, Gym, AER, and dance	20–40 min/week, 12 weeks total	-	-
Dâmaso et al. 2006 [44]	Brazil	Pre-post intervention	15–18	28	Obese	In PAL, RS	120 min/week, 12 weeks total	Diet intervention	Nutritional counselling every 3 weeks with a specialist
Farias et al. 2009 [45]	Brazil	Non-RCT	10–15	383	General	In school, monitored physical activity	120 min/week, 48 weeks total	-	-
Kain et al. 2008 [46]	Chile	Non-RCT	6–12	2039	General	In school, SW	90 min/week, 44 weeks total	Nutritional education	Educational material provided to students, teachers, and guardians
Kain et al 2009 [47]	Chile	Pre-post intervention	4–9	522	General	In school, SW	Not specified	Nutritional education	Educational materials provided to students and teachers
Lobos et al. 2013 [48]	Chile	Pre-post intervention	4–11	796	General	In school, physical education classes	240 min/week, 72 weeks total	Nutritional education	Educational materials about how to select healthy foods
Macias et al. 2010 [49]	Mexico	RCT	6–9	76	General	In school, LA	180 min/week, 12 weeks total	-	-
Marín et al. 2011 [50]	Chile	Pre-post intervention	6–18	268	Obese	In school, physical activity, AER, ANAER	60 min/week, 16 weeks total	Nutritional education	Handing out educational material for students and guardians
Patiño et al. 2013 [51]	Colombia	Pre-post intervention	7–11	9	Obese	Health clinic, AER, S	270 min/week, 12 weeks total	-	-
Perichart et al. 2007 [52]	Mexico	Pre-post intervention	8–14	360	General	In school, exercises: S, FLEX, B, and relaxation	100 min/week, 16 weeks total	Nutritional counselling	Messages during class time
Prado et al. 2009 [53]	Brazil	RCT	8–12	48	Obese	In PAL, AER, and G	60 min/week, 16 weeks total	-	-
Preciado et al. 2016 [54]	Colombia	Non-RCT	6–17	1003	General	In school, beginners’ sports	240 min/week, 8 weeks total	-	-
Ratner et al. 2013 [55]	Chile	Pre-post intervention	6–9	2527	General	In school, games and dance	90 min/week, 40 weeks total	Nutritional and diet education	Educational meetings about breakfast, nutritional labels, and snacks, among others
Shamah et al. 2012 [56]	Mexico	RCT	10–14	997	Overweight and obese	In school, games during recess	30 min/week 24 weeks total	Diet intervention	Lowering the caloric content of breakfast and including fruits and vegetables
Stabelini et al. 2016 [57]	Brazil	RCT	8–10	19	Obese	Counselling, increasing the number of steps taken every other day, and games	Every day, 12 weeks total	-	-
Vásquez et al. 2013 [58]	Chile	Pre-post intervention	8–13	120	Obese	In school, strength training	135 min/week, 12 weeks total	Educational group meetings	Information about nutrition and healthy diet for students and guardians

PAL: Physical Activity Laboratory; RS: Recreational Sports; S: Strength; FLEX: Flexibility; B: Balance; SW: Sports Workshop; OPEC: Optimised Physical Education Classes; AER: Aerobics; ANAER: Anaerobic; UNILAB: University Laboratory; G: Games; LA: Leisure Activities.

**Table 3 nutrients-12-01378-t003:** Quality Assessment.

COCHRANE COLLABORATIONS	Selection Bias	Performance Bias	Detection Bias	Attrition Bias	Reporting Bias	Other Bias	Risk of Bias
Alves et al. 2008 [41]	Low	Low	Low	Low	Unclear	Unclear	Low
Alvirde et al. 2013 [42]	High	Unclear	Low	High	Unclear	Unclear	High
Macías et al. 2009 [49]	Unclear	Unclear	High	Low	Unclear	Unclear	High
Prado et al. 2009 [53]	Unclear	Low	Unclear	Unclear	Unclear	Unclear	High
Shamah et al. 2012 [56]	Low	Low	Low	Low	Unclear	Unclear	Low
Stabelini et al. 2016 [57]	Unclear	Low	Unclear	Unclear	Unclear	Unclear	High
**EPHPP**	**Selection bias**	**Study design**	**Confounders**	**blinding**	**Data collection**	**Withdrawals/drop-outs**	**Risk of bias**
Balas et al. 2010 [43]	Moderate	Weak	Moderate	Weak	Strong	Moderate	Moderate
Dâmaso et al. 2006 [44]	Moderate	Moderate	Strong	Moderate	Strong	Moderate	Moderate
Farias et al. 2009 [45]	Low	Strong	Moderate	Moderate	Low	Moderate	Moderate
Kain et al. 2008 [46]	Low	Moderate	Moderate	Low	Moderate	Moderate	Moderate
Kain et al. 2009 [47]	Strong	Moderate	Moderate	Strong	Moderate	Strong	Moderate
Lobos et al. 2013 [48]	Moderate	Weak	Strong	Moderate	Moderate	Moderate	Moderate
Marín et al. 2011 [50]	Strong	Moderate	Moderate	Moderate	Moderate	Moderate	Moderate
Patiño et al. 2013 [51]	Weak	Weak	Moderate	Moderate	Moderate	Weak	Weak
Perichart et al. 2008 [52]	Moderate	Weak	Weak	Moderate	Moderate	Moderate	Moderate
Preciado et al. 2016 [54]	Moderate	Weak	Moderate	Moderate	Strong	Moderate	Moderate
Ratner et al. 2013 [55]	Moderate	Moderate	Strong	Strong	Strong	Moderate	Moderate
Vásquez et al. 2013 [58]	Strong	Moderate	Moderate	Strong	Strong	Moderate	Moderate

**Table 4 nutrients-12-01378-t004:** Sub-group analyses for study design, sex, and weight status.

	Physical Activity Only				Physical Activity Plus Diet			
Subgroup	*n*	ES	I2	*p*	*n*	ES	I2	*p*
		(95%CI)				(95%CI)		
**Intervention versus control effect**								
**Study design**								
RCT	3	0.05	40.8	0.185	4	−0.38	36.3	0.194
		(−0.36, 0.46)				(−0.50, −0.26)		
Non-RCT	3	−0.03	0.0	0.902	2	−0.11	0.0	0.887
		(−0.23, 0.17)				(−0.21, −0.01)		
**Sex**								
Boys	1	−0.02	-	-	2	−0.26	88.5	0.003
		(−0.30, 0.25)				(−0.59, 0.06)		
Girls	1	−0.05	-	-	2	−0.30	90.1	0.002
		(−0.34, 0.25)				(−0.67, 0.07)		
Both	3	0.05	40.8	0.185	2	−0.28	0.0	0.991
		(−0.36, 0.46)				(−0.40, −0.16)		
**Weight status**								
Overweight/obese	2	−0.17	0.0	0.758	3	−0.46	0.0	0.772
		(−0.57, 0.23)				(−0.58, −0.33)		
General population	3	0.05	27.9	0.250	3	−0.17	56.9	0.098
		(−0.17, 0.27)				(−0.29, −0.06)		
**Pre-post intervention effect**								
**Study design**								
RCT	4	−1.05	95.1	<0.001	3	−0.46	96.5	<0.001
		(−2.33, 0.24)				(−0.90, −0.01)		
Non-RCT	4	−0.04	0.0	0.552	2	0.22	0.0	0.859
		(−0.15, 0.07)				(0.15, 0.29)		
Single-arm pre-post study	3	−0.01	0.0	0.948	14	−0.38	95.5	<0.001
		(−0.22, 0.21)				(−0.64, −0.11)		
**Sex**								
Boys	2	−0.07	0.0	0.451	6	−0.09	94.8	<0.001
		(−0.19, 0.05)				(−0.39, 0.21)		
Girls	2	0.10	0.0	0.756	6	−0.04	92.9	<0.001
		(−0.15, 0.35)				(−0.30, 0.23)		
Both	7	−0.49	90.4	<0.001	7	−0.80	97.5	<0.001
		(−1.06, 0.08)				(−1.28, −0.33)		
**Weight status**								
Overweight/obese	3	−1.20	94.4	<0.001	7	−1.16	97.2	<0.001
		(−2.89, 0.49)				(−1.80, −0.52)		
General population	7	−0.01	0.0	0.729	12	0.08	76.3	<0.001
		(−0.11, 0.08)				(−0.01, 0.16)		

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
