# Peer review of "The Effects of Physical Activity and Diet Interventions on Body Mass Index in Latin American Children and Adolescents: A Systematic Review and Meta-Analysis"

_nutrients, 2020, doi:10.3390/nu12051378_

Round 1
Reviewer 1 Report
Introduction: Here you mention that there have been 4 systematic reviews on this topic in Latin American children, and there is an absence of meta-analyses, which is the rationale for your study. You should also mention in the introduction if there are existing meta-analyses of the effects of physical activity versus physical activity + diet on BMI in children in general (there are some) and what the results are.
Were there any studies that compared the three groups within the same study that could be mentioned in the discussion? i.e. did any study randomize children to 1) physical activity only, 2) physical activity + diet, and 3) control?
Given that these studies were in children, I wonder why you are assessing change in BMI as the primary outcome, rather than the change in BMI z-score. This is especially important for pre-post design studies. Children are growing in this time period, and their BMI is increasing through normal growth and maturation, so if you rely on BMI only, then you will underestimate the role of physical activity, especially in studies without a control group. This is a major issue that needs to be addressed.
I just read through the paper again, had only the three specific comments:
1) to add something about the results of existing meta-analyses in general in
the introduction (i.e. not just those in Latin America, 2) to add any
information about studies that have addressed this issue in their own study
design (i.e. physical activity versus physical activity + diet versus control,
and 3) to explain why they did not use BMI Z-score as the outcome in this
meta-analysis - they need to defend this approach with the rationale of why
they didn’t do this, and what the implications are for the results.
Author Response
Reviewer 1
Introduction: Here you mention that there have been 4 systematic reviews on this topic in Latin American children, and there is an absence of meta-analyses, which is the rationale for your study. You should also mention in the introduction if there are existing meta-analyses of the effects of physical activity versus physical activity + diet on BMI in children in general (there are some) and what the results are. Were there any studies that compared the three groups within the same study that could be mentioned in the discussion? i.e. did any study randomize children to 1) physical activity only, 2) physical activity + diet, and 3) control?
Given that these studies were in children, I wonder why you are assessing change in BMI as the primary outcome, rather than the change in BMI z-score. This is especially important for pre-post design studies. Children are growing in this time period, and their BMI is increasing through normal growth and maturation, so if you rely on BMI only, then you will underestimate the role of physical activity, especially in studies without a control group. This is a major issue that needs to be addressed.
I just read through the paper again, had only the three specific comments:
Comments 1: to add something about the results of existing meta-analyses in general in
the introduction (i.e. not just those in Latin America.
Authors: Thank you for the reviewer’s comment. As suggested, we have included a new sentence in introduction section as follows: included the following information
Lines 61 to 65.
“Previous studies analyzed the effect of physical activity interventions versus interventions that combine physical activity plus diet on BMI, showing a higher effect in the physical activity plus diet interventions groups [27,28]. In Latin America, four previous systematic reviews on interventions aimed at increasing physical activity in children have been performed, some of which were combined with diet intervention [29-32].”
Comments 2: to add any information about studies that have addressed this issue in their own study design (i.e. physical activity versus physical activity + diet versus control.
Authors: Thank you for the reviewer’s comment. As suggested, we have included the following information:
Lines 67 to 69:
“However, none of the four previously mentioned reviews performed meta-analyses to compare physical activity versus physical activity plus diet versus control groups.”
Lines 250 and 252:
“as was done in three of the studies included in this meta-analysis, which concluded that subjects that included physical activity plus diet programs proved to be more efficient in decreasing BMI values in children and adolescents [42,53,56].”
Comments 3: to explain why they did not use BMI Z-score as the outcome in this meta-analysis - they need to defend this approach with the rationale of why they didn’t do this, and what the implications are for the results.
Authors: Thank you for the reviewer’s comment. We understand the importance of the BMI Z-score in children, however, in the present study it was not possible to evaluate the changes in this indicator, since only two RCT studies, one non-RCT and two pre-post intervention, used the BMI Z-score. The following information was included in the study limitations.
Lines 287 to 291:
“Fifth, given that a limited number of studies included in this meta-analysis assessed the BMI Z-score, it was not possible to perform analyses based on this indicator. Therefore, it is suggested that, in future studies performed in Latin American children, the BMI Z-score should be incorporated, as this indicator reflects the changes produced by growth and maturation in more detail.”.
Reviewer 2 Report
I read with interest this review that shows a clear effect of PA+diet interventions on BMI in children and a clear lack of effect of PA only interventions. I hope my comments below can help you strengthen the manuscript.
Abstract
Please include details on how many studies were found and analysed and how many per subgroup analysis.
Please rewords "there was a decrease in BMI, but it was not statistically significant", to something like "there was no evidence of a decrease" because the decrease either exists or not.
If the effect estimate is not standardised, please include units on the effect estimate.
Please add heterogeneity estimates after each ES.
Please add a comment on the main limitation of the review.
Please add a comment on the overall quality of the evidence.
Methods
Please explain why terms around adolescents were not included in your search strategy but your study title mentions that this review was in adolescents, too. You can consider it in the limitations section of the discussion.
The DerSimonian and Laird method is widely acceptable but I would suggest considering the REML method as a sensitivity analysis. See https://www.ncbi.nlm.nih.gov/pubmed/30067315
Please clarify if your subgroup and sensitivity analyses were defined a priori or post-hoc.
Discussion
Please use first-person language throughout the paper, by revising phrases to "children with overweight and obesity".
Lines 263-265 are unclear, consider revising.
Another limitation to discuss is the publication bias you found.
I would consider changing the conclusion here and in the abstract to something like "PA+diet interventions produced a reduction in BMI but PA only interventions did not".
it will be useful to discuss how size of the effect in the interventions here compares with that of interventions in non-latin america settings and if there is a difference why this might be.
it would also be useful to see a discussion about how clinically meaningful such a "statistically significant" reduction in BMI with diet+PA is.
Line 270: "Despite" should be reworded to "based on"
Author Response
Reviewer 2
I read with interest this review that shows a clear effect of PA+diet interventions on BMI in children and a clear lack of effect of PA only interventions. I hope my comments below can help you strengthen the manuscript.
Abstract
Comments 1: Please include details on how many studies were found and analysed and how many per subgroup analysis.
Authors: Thank you for the reviewer’s comment. Done.
Comments 2: Please rewords "there was a decrease in BMI, but it was not statistically significant", to something like "there was no evidence of a decrease" because the decrease either exists or not.
Authors: Thank you for the reviewer’s comment. As suggested, we have rephrased as follows:
Lines 29 and 30:
“When ES was estimated considering only the effect in intervention groups, there was no evidence of a decrease in BMI …”
Comments 3: If the effect estimate is not standardised, please include units on the effect estimate.
Authors: Thank you for the reviewer’s comment. Pooled standardized mean difference for BMI was used as effect size estimate, thus this information has been included in the abstract.
Comments 4: Please add heterogeneity estimates after each ES.
Authors: Thank you for the reviewer’s comment. Done.
Comments 5: Please add a comment on the main limitation of the review.
Authors: Thank you for the reviewer’s comment. Done.
Lines 35 to 37:
“Some limitations of this review could compromise our results, but the main limitation that should be stated is the quality of the studies (mainly medium/moderate); especially as physical activity and diet interventions cannot be blinded, compromising the quality of these studies.”
Comments 6: Please add a comment on the overall quality of the evidence.
Authors: Thank you for the reviewer’s comment. Done.
Methods
Comments 7: Please explain why terms around adolescents were not included in your search strategy but your study title mentions that this review was in adolescents, too. You can consider it in the limitations section of the discussion.
Authors: The reviewer´s comment seems judicious. In order to avoid this limitation, we have modified the search strategy, including terms for adolescents. Figure 1 have been modified according to this new search strategy.
Comments 8: The DerSimonian and Laird method is widely acceptable but I would suggest considering the REML method as a sensitivity analysis. See https://www.ncbi.nlm.nih.gov/pubmed/30067315
Authors: The reviewer´s comment seems judicious. As suggested, we have included REML as a sensitivity analysis. We have included the following information:
Lines 143 and 144:
“Furthermore, all analyses were performed using the restricted maximum likelihood (REML) method to estimate the heterogeneity variance [39]”
Lines 200 to 202:
“After using REML method, heterogeneity was not modified for physical activity interventions versus control (I2 = 0.0%; p = 0.443) and slightly modified for physical activity plus diet versus control (I2 = 76.7%; p = 0.001).”
Lines 208 to 210:
“After using the REML method, heterogeneity was rather modified for physical activity interventions versus control (I2 = 98.0%; p < 0.001) and slightly modified for physical activity plus diet versus control (I2 = 99.2%; p < 0.001).”
Comments 9: Please clarify if your subgroup and sensitivity analyses were defined a priori or post-hoc.
Authors: Thank you for the reviewer’s comment. Done.
Line 149 and 150:
“Both analyzes were defined as post-hoc.”
Discussion
Comments 10: Please use first-person language throughout the paper, by revising phrases to "children with overweight and obesity".
Authors: Thank you for the reviewer’s comment. As suggested, we have used first-person language throughout the paper
Comments 11: Lines 263-265 are unclear, consider revising.
Authors: Thank you for the reviewer’s comment. We have rephrased the sentence as follows:
Lines 284 to 287:
“Fourth, the studies did not assess the daily physical activity of subjects performed outside of the interventions (either by questionnaire or accelerometer); thus, this confounding effect outside of the interventions could not be controlled.”
Comments 12: Another limitation to discuss is the publication bias you found.
Authors: Thank you for the reviewer’s comment. We have added publication bias in discussion.
Lines 291 and 292:
“Sixth, there was publication bias on physical activity plus diet pre-post intervention effects, due to the lack of studies with low ES and high sample size.”
Comments 13: I would consider changing the conclusion here and in the abstract to something like "PA+diet interventions produced a reduction in BMI but PA only interventions did not".
Authors: Thank you for the reviewer’s comment. As suggested, we have rephrased the conclusions as follows:
In abstract, lines 33 to 35:
“This meta-analysis offers evidence that physical activity plus diet interventions produced a reduction in BMI in Latin American children and adolescents, but physical activity only interventions did not.”
In conclusion, lines 294 and 295:
“To summarise, this meta-analysis offers evidence that physical activity plus diet interventions produced a reduction in BMI, but physical activity only interventions did not.”
Comments 14: it will be useful to discuss how size of the effect in the interventions here compares with that of interventions in non-latin america settings and if there is a difference why this might be.
Authors: Thank you for the reviewer’s comment.
Lines 236 to 239:
“Regarding the effect of physical activity plus diet interventions in the treatment of overweightness and obesity, our findings agree with previous investigations that studied non-Latin American populations, which reported an association between physical activity and BMI, as well as such interventions being an efficient way of lowering the percentage of adipose tissue [24,59-63]”
Lines 245 to 249:
“Our results on the effect of physical activity plus diet agree with previous meta-analyses that have analyzed non-Latin American populations [65,66], highlighting that physical activity is one of the central elements of weight loss; however, when combined with diet intervention, the reduction ranged from 3.2% to 20% more, underscoring that the best results are achieved when calories are restricted.”
Comments 15: it would also be useful to see a discussion about how clinically meaningful such a "statistically significant" reduction in BMI with diet+PA is.
Authors: Thank you for the reviewer’s comment.
Lines 253 to 257:
“The significant results found in the reduction of BMI by combining physical activity with diet are important from a clinical point of view, as evidence has been provided that will help to prevent and treat overweightness and obesity in children and adolescents, and which may also reduce the risk of acquiring metabolic and cardiovascular diseases, consequently reducing the economic expenditure on health that is currently generated by childhood overweightness and obesity [67].”
Comments 16: Line 270: "Despite" should be reworded to "based on"
Authors: Thank you for the reviewer’s comment. Done.
Round 2
Reviewer 1 Report
No further comments.
Author Response
Thank you for the reviewer’s comment.
Reviewer 2 Report
I am generally happy that my comments were addressed, but I would like clarity on one.
Following my comments, you mention that you have added "adolescents" in your search strategy. That's good but it is unclear if this was acted upon. Given your rapid response, I wasn't sure if you had re-run the search and screening with the new search strategy to ensure that no new studies had to be included. Can you please confirm that you have done so.
In your added text, you should revise overweightness to overweight.
In your abstract you should mention the conclusion after the limitations.
In line 143 where you describe the REML, you should mention for clarity that this was a sensitivity analysis. I also suggest you add the new REML estimates with 95% CIs in your results.
Author Response
Reviewer 2
I am generally happy that my comments were addressed, but I would like clarity on one.
Comments 1: Following my comments, you mention that you have added "adolescents" in your search strategy. That's good but it is unclear if this was acted upon. Given your rapid response, I wasn't sure if you had re-run the search and screening with the new search strategy to ensure that no new studies had to be included. Can you please confirm that you have done so.
Authors: The reviewer’s comment seems judicious. We confirm that we ran the search strategy again. Although the databases offered a greater number of articles after the application of the new search strategy, none of them met the inclusion criteria of this meta-analysis systematic review, so the number of included studies was not modified.
Comments 2: In your added text, you should revise overweightness to overweight.
Authors: Thank you for the reviewer’s comment. As suggested, “overweightness” has been replaced by “overweight”.
Comments 3: In your abstract you should mention the conclusion after the limitations.
Authors: Thank you for the reviewer’s comment. Done.
Comments 4: In line 143 where you describe the REML, you should mention for clarity that this was a sensitivity analysis. I also suggest you add the new REML estimates with 95% CIs in your results.
Authors: Thank you for the reviewer’s comment. As suggested, we have included the following information:
Lines 142 and 143:
“Furthermore, a sensitivity analyses were performed using the restricted maximum likelihood (REML) method to estimate the heterogeneity variance [39].”
Lines 199 to 202:
“After using REML method, ES and heterogeneity was not modified for physical activity interventions versus control (ES = 0.00; 95% CI -0.17–0.17; I2 = 0.0%; p = 0.443) and slightly modified for physical activity plus diet versus control (ES = 0.28; 95% CI -0.42– -0.13; I2 = 76.7%; p = 0.001).”
Lines 210 to 213:
“After using the REML method, ES and heterogeneity was rather modified for physical activity interventions versus control (ES= -0.36; 95% CI -1.07–0.36; I2 = 98.0%; p < 0.001) and slightly modified for physical activity plus diet versus control (ES= -0.37; 95% CI -0.79–0.05; I2 = 99.2%; p < 0.001).”